# A combined monthly precipitation prediction method based on CEEMD and improved LSTM

**Xinyun Jiang**●*

College of Information and Intelligence, Hunan Agricultural University, Changsha Hunan, PR China

* leiyun28@stu.hunau.edu.cn

## Abstract

With the continuous decline of water resources due to population growth and rapid economic development, precipitation prediction plays an important role in the rational allocation of global water resources. To address the non-linearity and non-stationarity of monthly precipitation, a combined prediction method based on complementary ensemble empirical mode decomposition (CEEMD) and a modified long short-term memory (LSTM) neural network was proposed. Firstly, the CEEMD method was used to decompose the monthly precipitation series into a set of relatively stationary sub-sequence components, which can better reflect the local characteristics of the sequence and further understand the nonlinear dynamic characteristics of the sequence. Then, improved LSTM neural networks were employed to predict each sub-sequence. The proposed improvement method optimized the hyper-parameters of LSTM neural networks using particle swarm optimization algorithm, which avoided the randomness of artificial parameter selection. Finally, the predicted results of each component were superimposed to obtain the final prediction result. The proposed method was validated by taking the monthly precipitation data from 1961 to 2020 in Changde City, Hunan Province as an example. The results of the case study show that, compared with other traditional prediction methods, the proposed method can better reflect the trend of precipitation changes and has higher prediction accuracy.

## Introduction

The World Meteorological Organization (WMO) published the report "State of Climate Services 2021: Water" in October 2021, stating that climate change, particularly the increased frequency of extreme heat, will cause a global water crisis and that governments and relevant international organizations are still lacking effective response measures [1]. Global heat, drought, and various types of extreme weather have resulted in a drop in global precipitation, yet with the rapid expansion of society, the usage of freshwater resources has increased more than 35 times in the previous 300 years, resulting in a water crisis in many parts of the world. In response, China's Ministry of Water Resources issued its 2022 Water Resources Management Essentials, which suggests increasing the level of refinement in water resource management. Precipitation forecasting is a key foundation for enhanced water resource management and an important instrument for extreme weather warnings. This research proposes a combined monthly precipitation prediction model based on CEEMD-PSO-LSTM to increase the

the following email: lal1112023@163.com. This is the email address of data administrator, Changde Meteorological Bureau, Hunan Province.

**Funding:** The author(s) received no specific funding for this work.

**Competing interests:** The authors have declared that no competing interests exist.

accuracy of monthly precipitation prediction and can provide a decision basis for energy warning and meteorological catastrophe prevention.

Numerous techniques for predicting precipitation have long been studied by researchers both domestically and overseas. The primary prediction techniques can be categorised into three groups: machine learning methods, statistical prediction methods, and causal prediction methods. The basic foundation of causal prediction methods is the study of climatic changes, atmospheric circulation processes, and the creation of causal models [2–4]. However, because there are so many variables that might affect atmospheric precipitation, choosing the right one makes it challenging to predict precipitation using the genesis model. To build prediction models based on historical precipitation data, statistical prediction methods use time-tested techniques like Markov chains [5] and autoregressive integrated moving average (ARIMA) [6]. The non-smooth and non-linear properties of precipitation data are frequently ignored by statistical prediction methods, which leads to typically subpar prediction accuracy. Due to the complexity and randomness of mid-long term precipitation, machine learning methods mostly use neural network algorithms to develop forecasting models, such as LSTM [7–9]. This can result in reduced convergence speed and forecasting accuracy of neural networks. In summary, single machine learning algorithms, statistical prediction models, and genetic prediction methods all have limitations and do not provide accurate prediction results for mid-long term precipitation. Both domestic and foreign scholars have made certain improvements to single neural network algorithms, such as optimizing the hyper-parameters of neural networks using optimization algorithms [10, 11], which avoids the randomness of manual selection. For nonlinear data, decomposition methods [9, 12, 13] can be used to decompose sequences into multiple IMFs in order to better capture the nonlinear components in the signal and improve prediction accuracy.

Empirical model decomposition (EMD) [14] is a decomposition model proposed by Huang N. E. in 1988 that has strong advantages for processing non-linear and non-stationary data. With the gradual deepening of EMD research, it is discovered that the process of adaptive decomposition in EMD inevitably involves modal confusion. Huang [15] et al. suggested the ensemble empirical model decomposition (EEMD) method, which successfully lessens the influence of the mode mixing phenomenon on the results while maintaining the physical uniqueness of the derived IMF components. In summary, the monthly precipitation data was decomposed by using the CEEMD method, which is an improvement over EEMD and effectively reduces the reconstructed signal noise so that the residual white noise is negligible;For nonlinear and non-stationary data, the signal characteristics are usually complex and dynamic, making it difficult to effectively process and extract them using traditional linear analysis methods such as Fourier analysis and wavelet analysis. However, the CEEMD algorithm based on EMD can achieve adaptive decomposition of non-linear and non-stationary signals and decompose them into multiple IMFs, thereby better capturing the non-stationary and non-linear components contained in the signal. LSTM neural networks are widely valued due to their superiority in capturing Spatio-temporal relationships [16]. In time series data, current data is often influenced by data from multiple previous time steps. Traditional feedforward neural networks cannot meet this requirement, resulting in errors in prediction. However, the design of long- and short-term memory units in LSTM networks allows for effective learning and memory of information from previous time steps, thereby better handling long-term dependencies and improving prediction performance. LSTM networks were used to predict the decomposed IMF components; to avoid the contingency of traditional manual selection of LSTM hyperparameters, the particle swarm algorithm [17–20] was used to find the optimal hyperparameters in LSTM. Compared with other optimization

algorithms, PSO (Particle Swarm Optimization) algorithm can quickly converge to the global optimal solution and usually has faster calculation speed than other algorithms. The PSO algorithm has good robustness and reliability, is less dependent on initial parameters, and is not easily trapped in local optimal solutions. A combined CEEMD-PSO-LSTM prediction model is established, which effectively improves the prediction accuracy on the basis of traditional precipitation prediction method.

To test the effectiveness of the model, Changde City, known as an international wetland city and an international garden city, was selected as the study area, and the model was applied to the monthly precipitation prediction of Changde City and compared with PSO-LSTM [21], EMD-LSTM [22] and EEMD-LSTM [23]. The analysis of the algorithm results shows that the combined CEEMD-PSO-LSTM prediction model has better prediction results and provides more effective decision support for water resources development planning for global development.

## Methodology

### CEEMD

Huang N E [14] proposed EMD in 1988 as part of the Hilbert-Huang transform (HHT), which has the advantage of processing non-stationary signals. The core of EMD is to decompose the signal into several intrinsic model function (IMF) and monotonic residual by using the signal polar point information. The EEMD decomposition is a solution to the problem of modal aliasing by adding normally distributed white noise to the original order and then decomposing the white noise as a whole to obtain the corresponding IMF components. In the CEEMD decomposition process, multiple sets of white noise with opposite signs are added to the original signal, which effectively reduces the noise of the reconstructed signal and achieves the goal of negligible residual white noise. The CEEMD decomposition process is as follows:

(1) Add $L$ sets of white noise of opposite sign to the original sequence $X(t)$ to obtain a positive noise sequence $X_1$ and a negative noise sequence $X_2$, with the total number of sequences being $2L$.

$$\begin{bmatrix} X_1 \\ X_2 \end{bmatrix} = \begin{bmatrix} 1 & 1 \\ 1 & -1 \end{bmatrix} \begin{bmatrix} X \\ N \end{bmatrix} \tag{1}$$

Where $N$ is the white noise sequence.

(2) The resulting sequence is decomposed by EMD separately to obtain $m$ IMF components; each group of components is noted as $IMF_{ij}^+(t)$ and $IMF_{ij}^-(t)$, where $i = 1, \ldots, L; j = 1, \ldots, m$.

(3) By taking the average of $IMF_{ij}^+(t)$ and $IMF_{ij}^-(t)$ for each set of IMF components, determine the value of the $j$ th component.

$$IMF_j = \frac{1}{2L} \sum_{i=1}^{L} \left[ IMF_{ij}^+(t) + IMF_{ij}^-(t) \right] \tag{2}$$

(4) Each set of IMF values obtained is accumulated to obtain the original sequence. The formula is as in Eq (3):

$$X(t) = \sum_{j=1}^{m} IMF_j(t) + r(t) \tag{3}$$

Where $r(t)$ is the residual trend term, i.e., the residual.

## PSO

The main idea of the PSO algorithm is to use "particles" to simulate the behaviour of a "flock of birds foraging". When PSO solves an optimisation problem, each particle has its own position, velocity, and fitness value. The problem solution corresponds to the position of each particle in the search space, and PSO completes the optimisation by the particle following the optimal solution it finds and the optimal solution of the population.

Assume that there is currently a $D$-dimensional search space with a population of $N$ particles, the position of the $i$ th particle is: $X_{id} = (x_{i1}, x_{i2}, \cdots, x_{iD})$; the velocity is: $V_{id} = (v_{i1}, v_{i2}, \cdots, v_{iD})$; the individual history optimal adaptation value is $f_p$; the population history optimal adaptation value is $f_g$; the individual optimal solution is $P_{id,\text{pbest}} = (p_{i1}, p_{i2}, \cdots, p_{iD})$; the population optimal solution is $P_{d,\text{gbest}} = (p_{1,\text{gbest}}, p_{2,\text{gbest}}, \cdots, p_{d,\text{gbest}})$. The velocity update equation is given by equation:

$$v_{id}^{k+1} = \omega v_{id}^k + c_1 r_1 (p_{id,\text{ phest}}^k - x_{id}^k) + c_2 r_2 (p_{d,\text{ gbest}}^k - x_{id}^k) \tag{4}$$

where $i$ is the particle serial number; $k$ is the number of iterations; $d$ is the particle dimension serial number; $\omega$ is inertia weight; $c_1$ and $c_2$ are the individual learning factor and population learning factor respectively; $r_1$ and $r_2$ are random numbers between 0 and 1; $v_{id}^k$ and $x_{id}^k$ are the velocity vector and position vector of particle $i$ in dimension $d$ at the $k$ th iteration respectively; $p_{id,\text{ phest}}^k$ is the historical optimal position of particle $i$ in dimension $d$ at the $k$ th iteration; $p_{d,\text{ gbest}}^k$ is the historical optimal position of the population in dimension $d$ at the $k$ th iteration; the first term to the right of the equal sign is the inertia term. The larger the $\omega$ the greater the inertial exploration capacity of the particle; the 2nd term to the right of the equal sign is the cognitive term, i.e. the distance and direction between the current position of a particle and its historically optimal position; the 3rd term to the right of the equal sign is the social term, i.e.the distance and direction between the current position of the particle and the optimal position of the group history. The position update equation is as follows:

$$x_{id}^{k+1} = x_{id}^k + v_{id}^{k+1} \tag{5}$$

If the current number of iterations reaches the pre-set maximum number or the minimum difference in the adaptation value between two iterations, stop the iteration and output the optimal solution, otherwise the particle velocity and position are updated using equations, and.

The MSE of LSTM network prediction was used as the particle fitness value:

$$\text{Fitness} = \frac{1}{n} \sum_{i=1}^{n} (y_i - \hat{y}_i)^2 \tag{6}$$

Where $y_i$ is the true value; $\hat{y}_i$ is the predicted value; $n$ is the number of samples.

## LSTM

The LSTM network is a special kind of recurrent neural network whose memory module structure retains the recurrent feedback mechanism while introducing gating units to control the rate of information accumulation and adding forgetting gates to selectively control the addition of new information, thus solving the long-term dependency problem that exists in sequence modelling.

The LSTM neural network adds input, output and forget gate structures to the RNN. The architecture of LSTM block is shown in Figs 1 and 2, which mainly consists of a memory unit, an oblivion gate, an input gate and an output gate, all of which take values in the range of (0, 1) and are controlled by a sigmoid function. The memory unit is the key component in the LSTM module, and its state at moment $t$ is $c_t$, which contains information about the long-term memory of the series. Assuming moment $t$, the inputs to the memory module in the LSTM include the input sequence $X_t$, state $c_{t-1}$ of memory cell at time $t-1$ and state $h_t$ of hidden layer at time $t-1$. In the memory module, the forgetting gate controls how much of the value of $c_{t-1}$ is forgotten at the previous moment and controls the degree of influence of $c_{t-1}$ on $c_t$; in the input gate, the linear combination of the state of the hidden layer at the previous moment $h_{t-1}$ and the input sequence $X_t$ are used as the input to the sigmoid function. $\tilde{c}_t$ is the

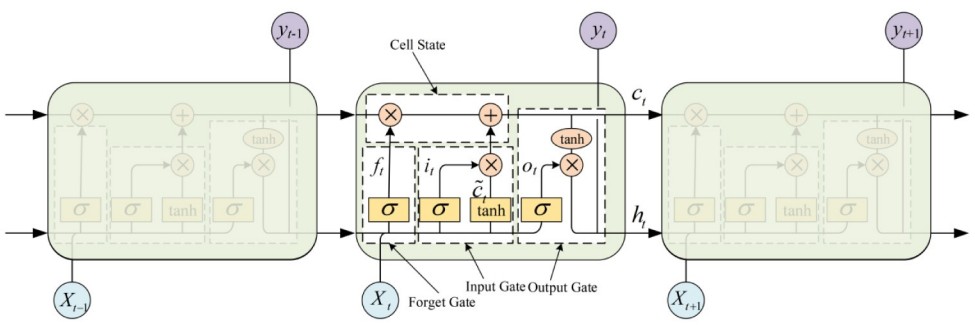

**Fig 1. Architecture of LSTM block.**

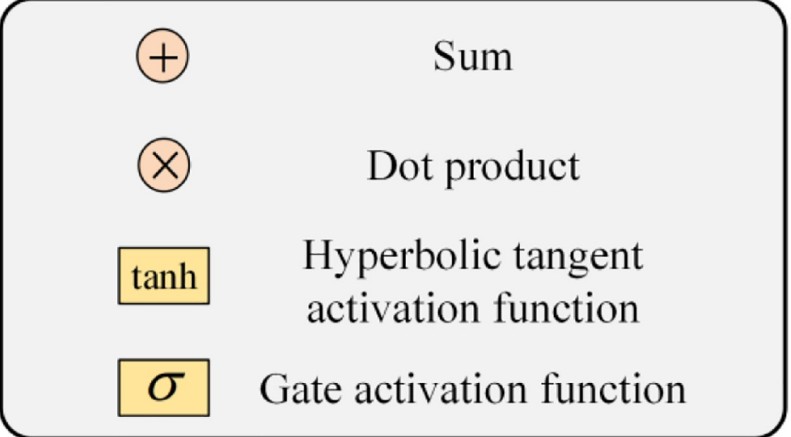

**Fig 2. Components of LSTM network.**

information retained after input gate control, i.e. the extent to which control $X_t$ affects $c_t$; in the output gate, the linear combination of the previous implicit state $h_{t-1}$ and the input sequence $X_t$ is used as input to the sigmoid function, which determines the output information to be retained by the memory neuron, i.e. the degree of influence of $c_t$ on $h_t$. Equations—for each of the three gates are as follows:

$$f_t = \sigma\left(W_f \cdot [h_{t-1}, X_t] + b_f\right) \tag{7}$$

$$i_t = \sigma\left(W_i \cdot [h_{t-1}, X_t] + b_i\right) \tag{8}$$

$$o_t = \sigma\left(W_o \cdot [h_{t-1}, X_t] + b_o\right) \tag{9}$$

where $f_t$, $i_t$, and $o_t$ are the output results of the forget, input, and output gates at time $t$; $W_f$, $W_i$, and $W_o$ are the weight matrices of the forget, input, and output gates; $b_f$, $b_i$, and $b_o$ are the bias terms of the forget, input, and output gates respectively; $\sigma$ is the sigmoid activation function.

The memory module $c_t$ is obtained by adding and multiplying $c_{t-1}$ with the addition of the information $\tilde{c}_t$ retained after being controlled by the input gate, calculated as follows:

$$\tilde{c}_t = \tanh(W_c \cdot [h_{t-1}, X_t] + b_c) \tag{10}$$

The state of the memory cell $c_t$ and the state of the hidden layer $h_t$ at time $t$ are calculated as follows:

$$c_t = f_t \cdot c_{t-1} + i_t \cdot \tilde{c}_t \tag{11}$$

$$h_t = o_t \cdot \tanh(c_t) \tag{12}$$

In Formula (10), $W_c$ and $b_c$ denote the weight matrix and bias term of the input cell state; tanh denotes the hyperbolic tangent activation function.

## Monthly precipitation prediction model based on CEEMD-PSO-LSTM

The predictability of monthly precipitation data using traditional prediction techniques is poor because they are inherently variable, non-linear, and non-smooth. This paper selects the CEEMD decomposition model and LSTM network prediction model, selects the PSO algorithm to optimise the hyperparameters in the LSTM, and proposes a combined CEEMD-PSO-LSTM prediction model in light of the benefits of empirical modal decomposition in series smoothing and the excellent performance of long and short-term memory neural networks in time series data prediction. The data decomposition, PSO optimization, and LSTM network prediction are the three key stages that make up the combined prediction model. Fig 3 illustrates the specific steps in this procedure.

(1) CEEMD decomposition phase. The CEEMD model is used to decompose the monthly precipitation data into the L-group *IMF* components {$IMF_1$, $IMF_2$, $\cdots$, $IMF_L$} and the residual error RES.

(2) PSO optimization phase. The PSO-LSTM model is used to make LSTM predictions for each set of IMF components separately, and the hyperparameters of the LSTM network are optimized by PSO. The specific steps are as follows:

① Initialize the particle swarm parameters. Determine the population size, number of layers, number of iterations, individual and population learning factors, limited range of particle position and velocity values, and inertia weights.

② Randomly initialise the particle velocity and position in the bounded range. Randomly generate a particle, $K$ is the number of iterations of the LSTM, $lr$ is the learning rate, $H_1$ is the number of neurons in layer 1 hidden layer, and $H_2$ is the number of neurons in layer 2 hidden layer.

③ Determine the fitness function of the PSO algorithm. The LSTM model is constructed with the initialized parameters, and the mean square error between the true value and the predicted value is used as the fitness function of the particle population, as shown in Fig 3.

④ Calculate the position and fitness value of the particles at each iteration. The positions and velocities of the particles are continuously adjusted according to equations until the fitness function is minimized to determine the optimal positions and thus the optimal parameters of the LSTM.

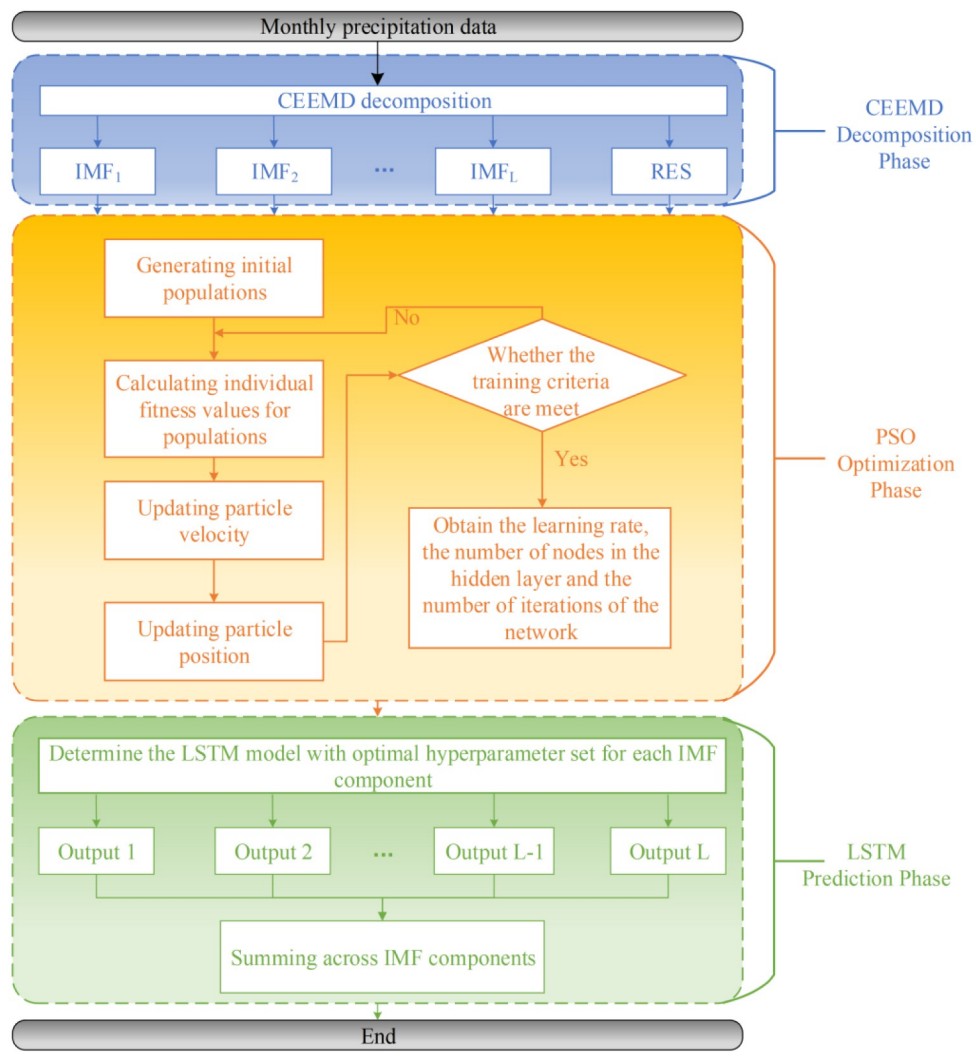

**Fig 3. Monthly precipitation data prediction process of CEEMD-PSO-LSTM combined model.**

(3) LSTM prediction phase. The optimal parameters determined after the PSO search are used to predict each set of IMF components obtained from the decomposition, and then the predicted values of each set of IMFs and the predicted values of the residual term RES are added together to obtain the final prediction results.

In this paper, RMSE and MAE are selected as evaluation indexes, and the specific formulas are as follows:

$$RMSE = \sqrt{\frac{\sum_{r=1}^{N} (x_r - \hat{x}_r)^2}{N}} \tag{13}$$

$$MAE = \frac{1}{N} \sum_{r=1}^{N} |x_r - \hat{x}_r| \tag{14}$$

$$MAPE = \frac{100\%}{n} \sum_{r=1}^{n} \left| \frac{\hat{x}_r - x_r}{x_r} \right| \tag{15}$$

In the formula: $x_r$ is the true value; $\hat{x}_r$ is the predicted value; $N$ is the number of samples.

## Experimental results

The experimental environment is Intel(R) Core (TM) i7-10510U, 2.30GHz processor, NVIDIA GeForce MX250 graphics card. Algorithm model uses MATLAB R2022a as programming language.

### Overview of the study area

Changde is in the south of mainland China, in the northwest of Hunan Province, and is famous as the "land of fish and rice" in the south of the Yangtze River. It is located in the Dongting Lake system in the middle reaches of the Yangtze River, the lower reaches of the Yuan River, and the middle and lower reaches of the river, as well as the northeastern end of the Wuling Mountains and the Xuefeng Mountains. Changde is 174.6 kilometres wide from east to west and 187.2 kilometres long from north to south, with a total area of 18,200 square kilometres. With an average annual temperature of 16.7˚C and 1200–1900 mm of precipitation, Changde has a subtropical humid monsoon climate. Water resources in Changde are relatively abundant, with a total of 15.337 billion cubic meters of water resources on average over the years, with a per capita possession of 2,556 cubic meters. Changde has abundant rainfall and water resources mainly come from precipitation, which is unevenly distributed in space and time, with precipitation and runoff accounting for more than 70% of the year during the period of abundant water (April to October). Changde is one of the first international wetland cities and international garden cities in China, so it is important to carry out medium and long-term precipitation forecasting work in the area.

### Research data sources and pre-processing

The monthly precipitation data from seven representative meteorological stations in Changde, Hunan Province, from January to December 1961 to 2020 were selected, and the distribution of meteorological stations is shown in Fig 4. In view of the limited space of the article, the validity and accuracy of the combined prediction model are mainly verified by using the measured data from Changde station with station number 57662.

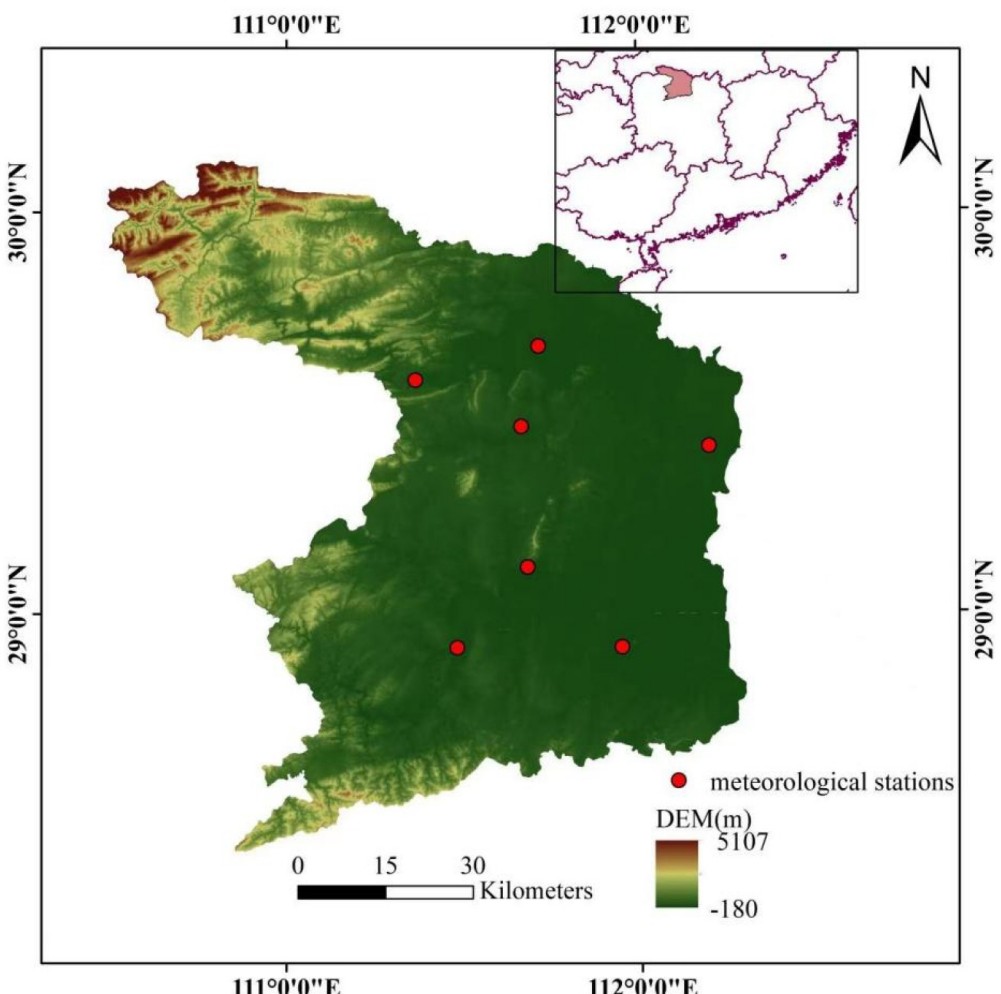

**Fig 4. Distribution of meteorological stations in Changde.**

The data is from Changde Meteorological Bureau. The data is true and accurate, part of the data is shown in Table 1. The number of data samples is 720.

Data for August to October 1976 were missing and Three times Hermite interpolation [24] were performed on the data for that year to maintain data continuity and reduce data loss. The final results obtained are shown in Table 2.

## CEEMD decomposition of monthly precipitation time series

The monthly precipitation time series has obvious non-linearity and non-smoothness, and CEEMD was used to decompose the series. When decomposed, add a white noise amplitude of 0.02 times the standard deviation of the original signal, set the average number of processing to 50. The original series was decomposed into eight IMF components [25], and the effect is shown in Fig 5, where each IMF presents the influence of different influencing factors on precipitation at different scales. Compared with the EMD IMFs, the CEEMD-processed IMFs do not show the mode mixing that often occurs in EMD, and each IMF contains significantly different characteristic time scales. After CEEMD processing, it can be seen that the auxiliary

**Table 1. Partial monthly precipitation in Changde from 1961 to 2020.**

| Year | Month | Precipitation | Year | Month | Precipitation |
|---|---|---|---|---|---|
| 1961 | 1 | 25.50 | 1993 | 1 | 80.20 |
| | 2 | 77.00 | | 2 | 125.70 |
| | …… | …… | | …… | …… |
| | 11 | 110.10 | | 11 | 73.70 |
| | 12 | 50.00 | | 12 | 28.50 |
| 1962 | 1 | 21.10 | 1994 | 1 | 46.00 |
| | 2 | 49.80 | | 2 | 55.90 |
| | …… | …… | | …… | …… |
| | 11 | 63.30 | | 11 | 67.00 |
| | 12 | 72.30 | | 12 | 54.10 |
| …… | …… | …… | …… | …… | …… |
| 1991 | 1 | 96.40 | 2019 | 1 | 54.00 |
| | 2 | 113.40 | | 2 | 71.30 |
| | …… | …… | | …… | …… |
| | 11 | 28.20 | | 11 | 57.50 |
| | 12 | 34.80 | | 12 | 18.40 |
| 1992 | 1 | 17.00 | 2020 | 1 | 107.80 |
| | 2 | 38.70 | | 2 | 68.60 |
| | …… | …… | | …… | …… |
| | 11 | 14.20 | | 11 | 80.40 |
| | 12 | 75.80 | | 12 | 15.20 |

noise residuals of the IMF components are decreased and the signal-to-noise ratio is increased compared to the EEMD IMFs. This allows the information of the original series to be more accurately reflected, and the total number of set averaging, the decomposition takes less time.

## PSO-LSTM network prediction

LSTM prediction was performed on each of the eight IMF and RES sequences obtained after CEEMD decomposition. Before making the predictions, each sequence data was first normalised separately with the following equation.

$$X_i = \frac{x_i - x_{\min}}{x_{\max} - x_{\min}} \qquad (16)$$

where $x_i$ is the original data; $x_{\max}$, $x_{\min}$ are the maximum and minimum values of the original data respectively; and $X_i$ is the normalised data.

The timestep of the input samples for the LSTM network is 12, with 12 consecutive months of data as the input variable and the next month's data as the output variable. The data set is divided as shown in Table 3, and the network uses a mini-batch input, with the number of samples per input, batch-size = 16.

**Table 2. Results of data preprocessing.**

| Month | 8 | 9 | 10 |
|---|---|---|---|
| Monthly precipitation(mm) | 95.80 | 104.07 | 122.25 |

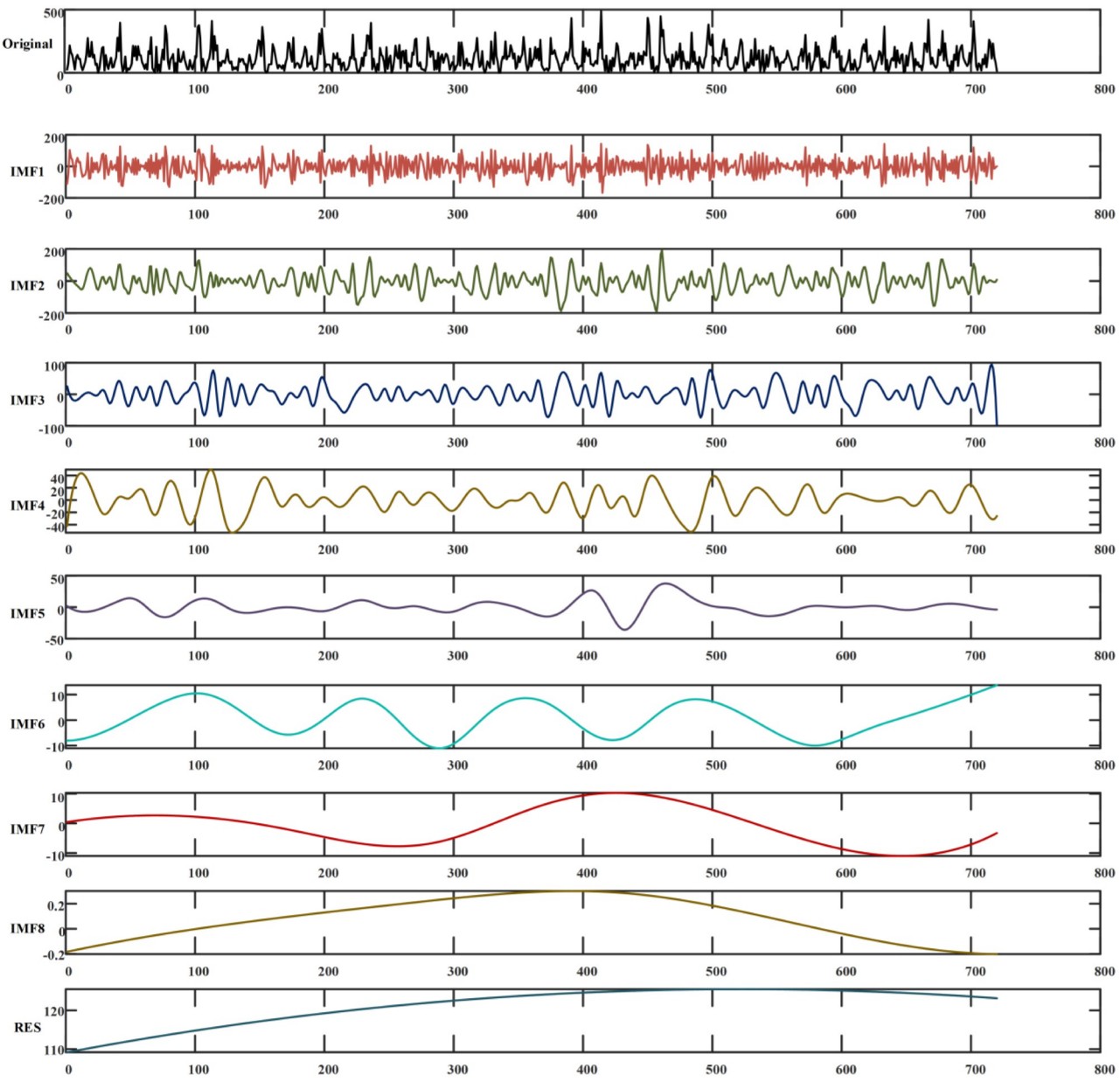

**Fig 5. CEEMD decomposition results of monthly precipitation.**

The LSTM network architecture adopts a 2 + 1 stack structure (2 layers of LSTM and 1 layer of fully connected layer). In order to prevent the neural network from overfitting, Dropout technology is added to each layer [26] (parameter value is 0.2). Dropout technology is to randomly discard some neurons in the neural network model, the weights of the discarded

**Table 3. Dataset partitioning and partial parameter settings.**

| Data set | Partition ratio | timestep | batch-size |
|---|---|---|---|
| Training set | 70% | 12 | 16 |
| Testing set | 30% | 12 | 16 |

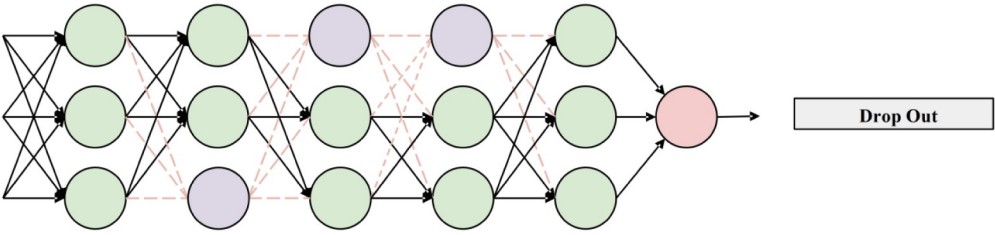

**Fig 6. Schematic of Dropout.**

neurons are set to zero, and discard neurons do not participate in network training forward calculation and reverse calculation, reducing the weight parameters, reduce the overfitting phenomenon, Fig 6 is a schematic diagram of Dropout technology. The neural network training uses the full iterative epoch method (the sample set is recorded as 1 epoch after training), the loss function uses the root mean square error, and the gradient optimization algorithm uses Adam [27].

After dividing the training set, the training set data is input into LSTM, and a loss value is generated by forward propagation calculation. According to the loss value, the Adam optimizer uses the BPTT algorithm to adjust the weight of LSTM. By comparing the fitness function, the PSO algorithm is used to find the optimal number of two hidden layer units $H_1$, $H_2$, the number of training $K$, and the learning rate $lr$. The PSO partial parameter settings are shown in Table 4, $M$ is the maximum iteration parameter. $H_1$, $H_2$ in the range [1, 200], $K$ in the range [10, 100], $lr$ in the range [0.001, 0.01]. As the number of training iterations increases, the accuracy of LSTM predictions improves. The results of hyper-parameter optimization can be found in Table 5. After LSTM training, the test set data is input into LSTM, the denormalized results are compared with the actual results, and the error indicators are used to evaluate the prediction performance of LSTM.

**Table 4. Some parameters of PSO.**

| $\omega$ | $c_1$ | $c_2p$ | $N$ | $M$ |
|---|---|---|---|---|
| 0.8 | 1.5 | 1.5 | 5 | 10 |

**Table 5. Some parameters of PSO.**

| Subsequence | $H_1$ | $H_2$ | $K$ | $lr$ |
|---|---|---|---|---|
| IMF1 | 144 | 148 | 78 | 0.0063 |
| IMF2 | 189 | 123 | 86 | 0.0075 |
| IMF3 | 173 | 107 | 95 | 0.0062 |
| IMF4 | 60 | 49 | 99 | 0.0097 |
| IMF5 | 141 | 178 | 68 | 0.0099 |
| IMF6 | 50 | 139 | 81 | 0.0070 |
| IMF7 | 68 | 82 | 69 | 0.0083 |
| IMF8 | 119 | 50 | 65 | 0.0053 |
| RES | 67 | 179 | 42 | 0.0060 |

## Simulation comparison results analysis

The above sub-series components predicted by the LSTM model were overlaid and reconstruction was carried out to obtain the monthly precipitation predictions. In this section, the research idea of verifying the superiority of the CEEMD-PSO-LSTM prediction model is mainly divided into three steps, and the evaluation indexes are RMSE and MAE. In the first step, BP neural network [28], SVM [29] and ANN [30] are used to compare the prediction accuracy of the three commonly used models with the LSTM model to prove the advantages of LSTM in dealing with time series modeling problems. In the second step, compare the EMD-LSTM, EEMD-LSTM and CEEMD-LSTM [31, 32] combined prediction models to prove the superiority of using CEEMD to process non-stationary data. In the third step, PSO-LSTM, CEEMD-LSTM and CEEMD-PSO-LSTM [33] models are compared to prove that the CEEMD-PSO-LSTM combined model has the best prediction performance.

The prediction results of the above prediction model in the test set samples are shown in Figs 7 and 8, and the predictors are shown in Table 6. Compared with BP and ANN, SVM and LSTM models have lower prediction error metrics and LSTM models have the highest prediction accuracy, with RMSE, MAE and MAPE of 77.92, 59.17 and 43.81%, respectively. Compared with ANN, RMSE, MAE and MAPE decreased by 27.9%, 26.76% and 21.85%, respectively. It can be seen from Fig 8 that the LSTM model can still have more accurate accuracy at the point where the monthly precipitation changes greatly, which proves that the LSTM model can capture the temporal correlation in the data more effectively. Compared with EMD-LSTM and EEMD-LSTM, RMSE and MAE of CEEMD-LSTM are lower than those of the other two models. As can be seen from Fig 8, although the trend of these three models is basically the same as the actual value, the CEEMD-LSTM predicted value is basically consistent with the actual value at the inflection point, which shows the superiority of the CEEMD decomposition method for processing non-stationary data. Finally, compared with PSO-LSTM and CEEMD-LSTM models, the RMSE of CEEMD-PSO-LSTM model is reduced by 25.26% and 6.43% respectively, and the MAE is reduced by 21.14% and 8.37% respectively. In Fig 8, where the sample points fluctuate greatly, the CEEMD-PSO-LSTM method has better prediction results, which proves that the CEEMD-PSO-LSTM method has better estimated performance. Through Figs 8 and 9, the CEEMD-PSO-LSTM method improves the overall forecasting precision while controlling the deviation of most prediction points from the actual data within a small range. This is due to the fact that the model decomposes the monthly precipitation data into a number of sub-series with significant regularity before forecasting them separately, improving the accuracy of the prediction.

## Conclusion

This paper combines the current research hotspots in the field of deep learning and focuses on the prediction accuracy of monthly precipitation to conduct research and establish a CEEMD-PSO-LSTM prediction model. Initially, the CEEMD decomposition algorithm was used to decompose the monthly precipitation with non-linearity and non-stationary characteristics into sub-sequences. Then, PSO was used to optimize the hyper-parameters of the LSTM network. Finally, the LSTM model was used to predict each sub-sequence, and the predicted results were combined to obtain the final monthly precipitation prediction. The following conclusions were drawn:

(1) The CEEMD decomposition method is used to decompose the monthly precipitation series, which reduces the interaction between different time scale information. Compared with EMD-LSTM and EEMD-LSTM methods, the RMSE of CEEMD-LSTM is reduced by

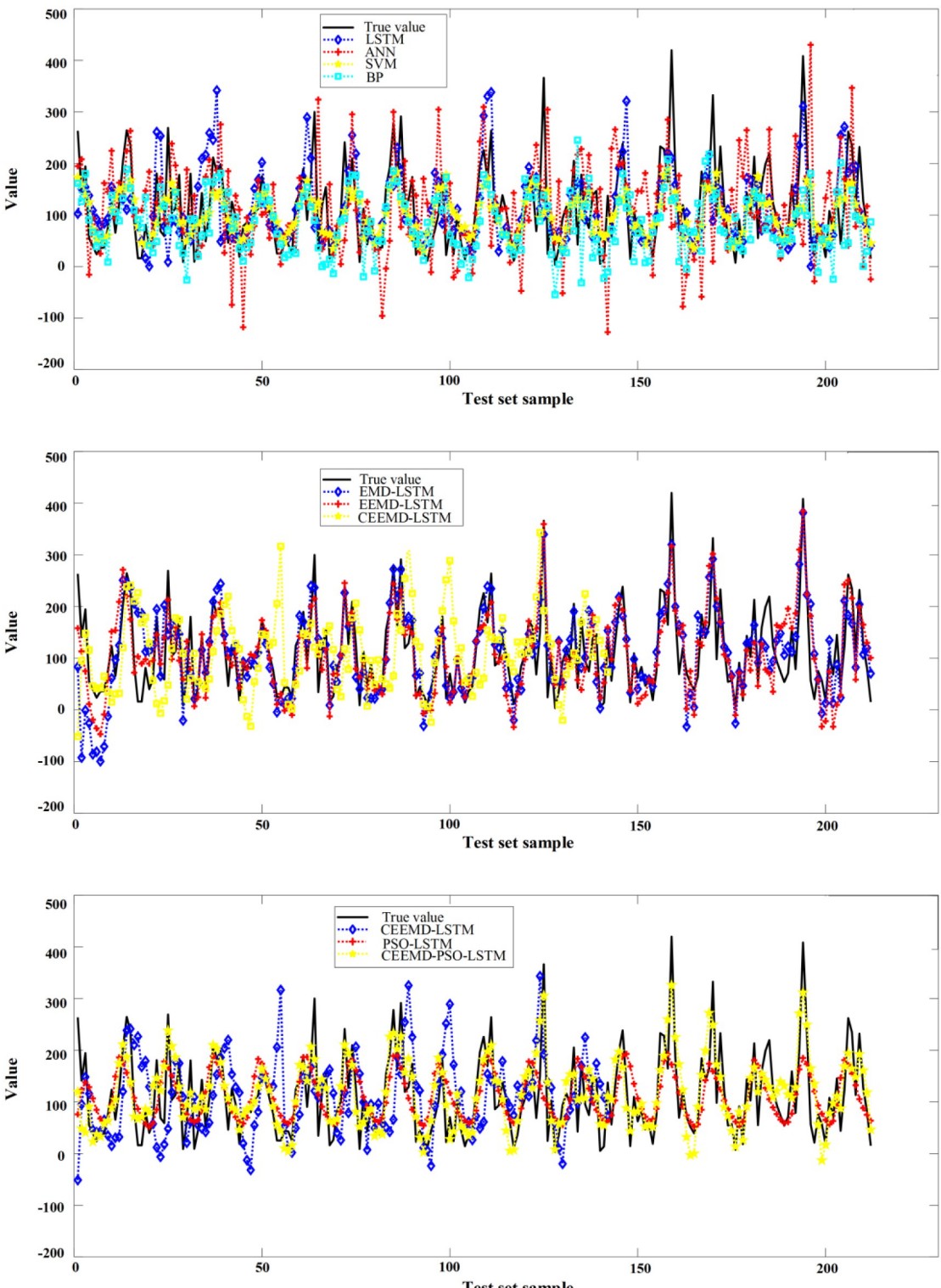

**Fig 7. Prediction results of the comparison models.**

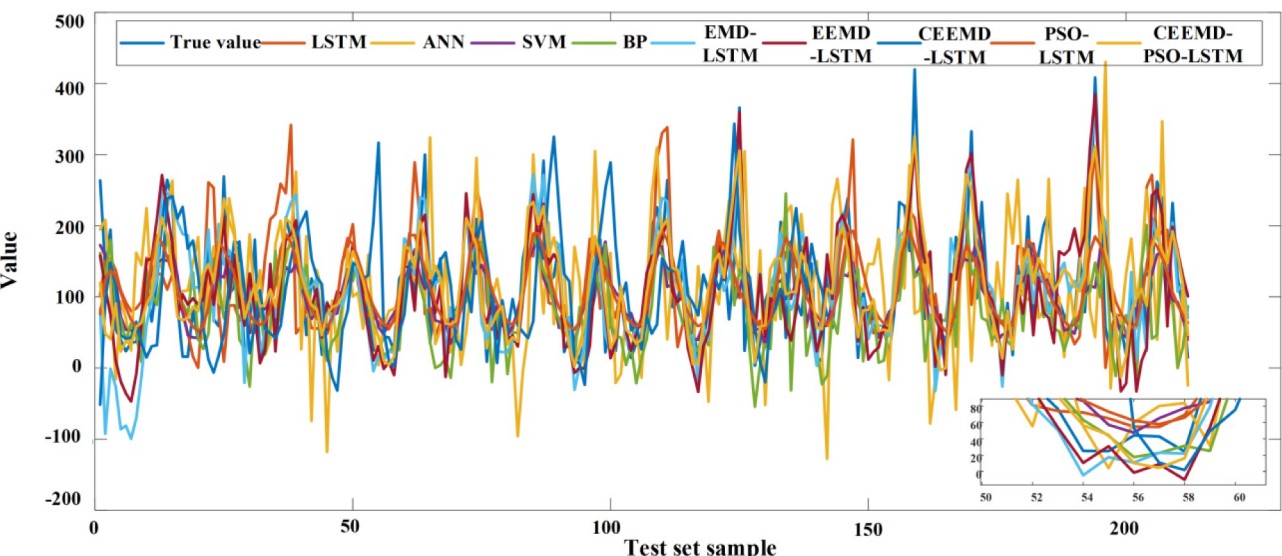

**Fig 8. Prediction results of the comparison models.**

13.78% and 3.75% respectively, MAE is reduced by 6.21% and 0.9% respectively, and MAPE is reduced by 13.83% and 6.95% respectively. It shows that the CEEMD effectively improves prediction accuracy.

(2) The PSO algorithm is used to optimize the hyper-parameters of the LSTM network, which avoids the contingency of manual selection. Compared with other optimization algorithms, PSO algorithm can quickly converge to the global optimal solution and have good robustness. LSTM network has certain superiority in handling time series data. Through the design of LSTM units, the LSTM network can effectively learn and remember information from previous time steps, thereby better handling long-term dependencies and improving prediction performance.

(3) The combined CEEMD-PSO-LSTM model has been developed to effectively improve the accuracy of monthly precipitation prediction. The model is suitable for processing non-smooth, non-linear time-series data and can also be extended to the fields of electricity, traffic flow and text recognition.

**Table 6. Some parameters of PSO.**

| prediction model | evaluation index | | |
|---|---|---|---|
| | RMSE | MAE | MAPE |
| LSTM | 77.92 | 59.17 | 43.81% |
| ANN | 108.08 | 80.79 | 56.06% |
| SVM | 79.08 | 60.96 | 53.38% |
| BP | 81.90 | 62.19 | 50.38% |
| EMD-LSTM | 60.76 | 45.84 | 27.34% |
| EEMD-LSTM | 54.43 | 43.40 | 25.32% |
| CEEMD-LSTM | 52.3923 | 42.99 | 23.56% |
| PSO-LSTM | 65.59 | 49.95 | 37.61% |
| CEEMD-PSO-LSTM | 49.02 | 39.39 | 20.65% |

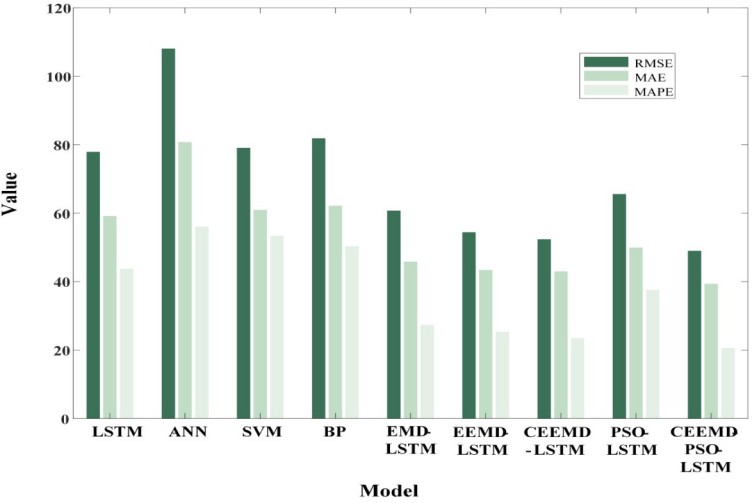

**Fig 9. Bar chart of the predication results.**

(4) In the next step, other influencing factors [34–37] can be introduced, such as pressure, temperature, etc., to further improve the reliability and prediction accuracy of the model.

## Author Contributions

**Conceptualization:** Xinyun Jiang.

**Data curation:** Xinyun Jiang.

**Formal analysis:** Xinyun Jiang.

**Funding acquisition:** Xinyun Jiang.

**Investigation:** Xinyun Jiang.

**Methodology:** Xinyun Jiang.

**Project administration:** Xinyun Jiang.

**Resources:** Xinyun Jiang.

**Software:** Xinyun Jiang.

**Supervision:** Xinyun Jiang.

**Validation:** Xinyun Jiang.

**Visualization:** Xinyun Jiang.

**Writing – original draft:** Xinyun Jiang.

**Writing – review & editing:** Xinyun Jiang.

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
