## [Decision Letter · Decision Letter 0]

16 May 2023

PONE-D-23-13513A combined monthly precipitation prediction method based on CEEMD and improved LSTMPLOS ONE

Dear Dr. jiang,

Thank you for submitting your manuscript to PLOS ONE. After careful consideration, we feel that it has merit but does not fully meet PLOS ONE’s publication criteria as it currently stands. Therefore, we invite you to submit a revised version of the manuscript that addresses the points raised during the review process.

ACADEMIC EDITOR: We kindly ask you to revise the paper considering the Reviewers' remarks and suggestions presented below. When this process is completed, the paper may be acceptable for publication.

We look forward to receiving your revised manuscript.

Kind regards,

Lin Wang, Ph.D.

Academic Editor

PLOS ONE

Journal Requirements:

Reviewers' comments:

Reviewer's Responses to Questions

**Comments to the Author**

1. Is the manuscript technically sound, and do the data support the conclusions?

Reviewer #1: Yes

Reviewer #2: Yes

2. Has the statistical analysis been performed appropriately and rigorously? 

Reviewer #1: Yes

Reviewer #2: Yes

3. Have the authors made all data underlying the findings in their manuscript fully available?

Reviewer #1: Yes

Reviewer #2: Yes

4. Is the manuscript presented in an intelligible fashion and written in standard English?

Reviewer #1: Yes

Reviewer #2: No

5. Review Comments to the Author

Reviewer #1: 1.Refine your innovation in Introduction Section.

2.Review of the literature can be improved further, may include the study of other applications of deep learning (10.1007/s10489-022-04254-0; https://doi.org/10.1016/j.energy.2022.123990; 10.1007/s00521-022-07967-y).

3.Add MAPE to evaluate these model.

4.Why use PSO instead of other methods to optimize LSTM?

5.The Conclusions have not been written well. It is not concise and clear. It is better to explain briefly the research gaps, research objectives, and methodology, and then clearly present the main results obtained in the work.

Reviewer #2: (a) Plz make Abstract informative and concise.

(b) The advantages of CEEMD should be further clarified.

(c) Section 3.2.3: Some studied of LSTM and applications should be added to highlight the advantages of LSTM (https://doi.org/10.1016/j.energy.2022.126100; Effective energy consumption forecasting using empirical wavelet transform and long short-term memory. Energy, 2022, 238: 121756; LSTM-Based Model Predictive Control of Piezoelectric Motion Stages for High-Speed Autofocus).

(d) To improve the practicality, how to set the related parameters of LSTM for better performance?

6. PLOS authors have the option to publish the peer review history of their article (what does this mean?). If published, this will include your full peer review and any attached files.

Reviewer #1: No

Reviewer #2: No

---

## [Author Response · Author response to Decision Letter 0]

15 Jun 2023

Thank you for your review comments. We appreciate your opinions and suggestions on our research. We have carefully read your review comments and made revisions and improvements to the article based on your suggestions.

We have conducted an in-depth analysis and discussion on the issues you raised, and have taken corresponding measures to revise and improve the article. We believe that after our efforts, the quality of the article has been significantly improved.

We greatly appreciate your review comments and assistance, which have been extremely valuable to our research. Thank you once again for your support and help!In the "Response to Reviewers" document, we have provided a detailed response to your suggestions.

---

## [Decision Letter · Decision Letter 1]

22 Jun 2023

A combined monthly precipitation prediction method based on CEEMD and improved LSTM

PONE-D-23-13513R1

Dear Dr. jiang,

We’re pleased to inform you that your manuscript has been judged scientifically suitable for publication and will be formally accepted for publication once it meets all outstanding technical requirements.

Kind regards,

Lin Wang, Ph.D.

Academic Editor

PLOS ONE

Additional Editor Comments (optional):

Reviewers' comments:

Reviewer's Responses to Questions

**Comments to the Author**

1. If the authors have adequately addressed your comments raised in a previous round of review and you feel that this manuscript is now acceptable for publication, you may indicate that here to bypass the “Comments to the Author” section, enter your conflict of interest statement in the “Confidential to Editor” section, and submit your "Accept" recommendation.

Reviewer #1: All comments have been addressed

Reviewer #2: All comments have been addressed

2. Is the manuscript technically sound, and do the data support the conclusions?

Reviewer #1: Yes

Reviewer #2: Yes

3. Has the statistical analysis been performed appropriately and rigorously? 

Reviewer #1: Yes

Reviewer #2: Yes

4. Have the authors made all data underlying the findings in their manuscript fully available?

Reviewer #1: Yes

Reviewer #2: Yes

5. Is the manuscript presented in an intelligible fashion and written in standard English?

Reviewer #1: Yes

Reviewer #2: Yes

6. Review Comments to the Author

Reviewer #1: This paper has been very well revised to my comments. The comments are at the level of Plos one publication and I recommend this paper for acceptance.

Reviewer #2: The revision is satisfactory. I can be accepted at the present version. No further comments to this paper.

7. PLOS authors have the option to publish the peer review history of their article (what does this mean?). If published, this will include your full peer review and any attached files.

Reviewer #1: No

Reviewer #2: No

---

## [Editor Report · Acceptance letter]

5 Jul 2023

PONE-D-23-13513R1 

A combined monthly precipitation prediction method based on CEEMD and improved LSTM 

Dear Dr. Jiang:

I'm pleased to inform you that your manuscript has been deemed suitable for publication in PLOS ONE. Congratulations! Your manuscript is now with our production department. 

Kind regards, 

on behalf of

Dr. Lin Wang 

Academic Editor

PLOS ONE